# The Influence and Optimization of Geometrical Parameters on Coast-Down Characteristics of Nuclear Reactor Coolant Pumps

**Yuanyuan Zhao, Xiangyu Si, Xiuli Wang * , Rongsheng Zhu , Qiang Fu and Huazhou Zhong**

National Research Center of Pumps, Jiangsu University, Zhenjiang 212013, China; zyy-michelle@163.com (Y.Z.); sixiangyu01@163.com (X.S.); ujs_zrs@163.com (R.Z.); ujsfq@sina.com (Q.F.); huazhou.zhong@turbotides.com.cn (H.Z.)

* Correspondence: ujswxl@ujs.edu.cn; Tel.: +86-1860-5110-959

**Abstract:** Coast-down characteristics are the crucial safety evaluation factors of nuclear reactor coolant pumps. The energy stored at the highest moment of inertia of the reactor coolant pump unit is utilized to maintain a normal coolant supply to the core of the cooling loop system for a short period of time during the coast-down transition. As a result of the high inertia moment of the rotor system, the unit requires a high reliability of the nuclear reactor coolant pump and consumes considerable energy in the start-up and normal operation. This paper considers the operational characteristics of the coast-down transition process based on the existing hydraulic model of the nuclear reactor coolant pump. With the implementation of an orthogonal test, the hydraulic performance of the nuclear reactor coolant pump was optimized, and the optimal combination of impeller geometrical parameters was selected using multivariate linear regression to prolong the coast-down time of the reactor coolant pump and to avoid serious nuclear accidents.

**Keywords:** reactor coolant pump; coast-down characteristics; geometrical parameters; multiple linear regression; transition process

---

## 1. Introduction

The nuclear reactor coolant pump is the only rotating piece of equipment in the primary loop cooling system, and thus can be called the 'heart' of the nuclear power plant. In the unfortunate event of a power failure at the nuclear power plant, the reactor coolant pump loses its power source and it will enter a coast-down state. For a short period of time, the inertia of the inert wheel provides power for the reactor coolant pump and the coolant continues to cool the reactor core. The coast-down process of the nuclear reactor coolant pump can be seen as a typical transient process. The nuclear reactor has a short coast-down transient process and cooling circuit working time. As the heat of the reactor cannot be discharged in a short time the temperature tends to rise sharply, leading to the potential decomposition of the coolant and generating a large amount of hydrogen, which is not at all conducive to the safety of the system [1]. Hence, the study of the effect of dynamic characteristics on the nuclear reactor coolant pump during the coast-down transient process becomes imperative.

Very limited literature is available related to the transient process of domestic and foreign nuclear reactor coolant pumps. Nevertheless, the transient processes of the centrifugal pumps and the mixed flow pumps have been widely studied. The transient characteristics of the centrifugal pump in its acceleration and deceleration process were determined by Tsukamoto et al. [2] using a theoretical analysis. Wu and Li et al. [3,4] examined the starting and stopping transients of the centrifugal pumps and mixed flow pumps by combining experimental and numerical calculations. A systematic study

on the start-up process of centrifugal pumps with different valve opening degrees, different impeller outside diameters, different blade widths, and different starting descending speeds was conducted by Elaoud et al. [5,6]. By establishing a mathematical start-up model for the primary circuit cooling system, Farhadi et al. [7–9] studied the effects of the ratio between the inertial energy of the unit and the fluid mass inertia energy of the pipe coolant during the start-up process of the nuclear reactor coolant pump. Whether the pumping capability meets the coast-down half time requirement as prescribed by safety analyses during the coast-down period was studied by Alatrash et al. [10] using experiments. Yonggang [11–13] studied the third and fourth generation nuclear main pumps, including gas–liquid two-phase flow and structural optimization. There have been numerous valuable studies on the coast-down characteristics of nuclear reactor coolant pumps, but comparatively little research details are available on the factors that affect the coast-down characteristics.

Geometric parameters of the impeller are one of the main factors affecting pump performance. Based on a numerical analysis of a 3D viscous flow, Hyuk et al. [14] designed a high-efficiency mixed-flow pump and the results suggested that the hydraulic efficiency of a mixed-flow pump at the design level can be improved by modifying its geometry. The impact of the geometry parameters of the impeller on the hydraulic performance of mixed flow pumps was studied by Varchola et al. [15], and several different designs were also compared. Long et al. [16] studied how the blade numbers of the impeller and the diffuser influence the reactor coolant pump performances using the numerical simulation method. The effect of the blade stacking lean angle on the hydraulic performance of a 1400 MW nuclear reactor coolant pump was studied by Zhou et al. [17], and it was determined that the geometric parameters such as the blade stacking lean angle highly influences the hydraulic efficiency of different flow intervals. Evidently, although the influence of the geometrical parameters on the pump has been studied, the research on the coupling effect of the nuclear reactor coolant pump has been very limited.

With regards to the factors influencing the coast-down characteristics of reactor coolant pumps, the indirect coupling effects between the different geometric parameters and combinations have been examined in depth by this paper. Changing the pump performance by changing the size of a certain geometric parameter virtually changes the direct impact of this parameter on performance, with an indirect impact on the performance of the other parameters simultaneously. This paper employs the multiple linear regression to analyze the optimal geometrical parameters of the impeller, based on the relationship between geometric parameters of the impeller and its efficiency, and the head.

## 2. Research Method

In the event of an unfortunate loss of power source to the nuclear reactor coolant pump, the unit utilizes its own moment of inertia to store energy to ultimately maintain the operation of the reactor coolant pump for a longer stretch of period, this phenomenon is known as the coast-down characteristic of the nuclear reactor coolant pumps [18]. In the coast-down transition process, the energy stored by the flywheel ensures that the time of the nuclear reactor coolant pump flow decreases by half within the specified safe time margin. The flywheel of an AP1000 nuclear reactor coolant pump is usually split into the upper flywheel and the lower flywheel while being fixed on the main shaft. To maximize improvement in the moments of inertia at a limited volume to maintain the coast-down characteristics of the reactor coolant pump, the flywheels were usually encompassed with high-density heavy metal tungsten alloy blocks and high-quality stainless-steel wheels. With respect to the issue of energy consumption, the main function of the flywheel was to provide energy to keep the nuclear reactor coolant pump running during the coast-down transition, and the flywheel should consume enough amounts of energy to maintain the start-up process and normal operation. It was suggested in the combination of Equations (1) and (2) that the time of coast-down was not only affected by the moment of inertia, but the efficiency of the rated operating point and the energy loss of the coast-down transition were also significant factors. As the rotational inertia of the impeller was obviously smaller than that of the flywheel, the moment of inertia of the rotor basically remains unchanged when the geometric

parameters of the impeller were changed [18–20]. Accordingly, this study endeavors to increase the efficiency of the hydraulic model rated point by optimizing the impeller geometric parameters to reduce any extra energy loss in the coast-down transition, to extend the time of coast-down transition, to increase the system reliability, and to reduce the cost thereon.

$$E_T = 2\pi^2 \int (\oint \rho n^2(t) A dz) dt + E_f \tag{1}$$

$$t = \frac{P_0}{4\pi^2 I \eta_0 n_0^2} [(\frac{n_0}{n(t)} - 1)] \tag{2}$$

In Equations (1) and (2), $E_T = \frac{1}{2} I_P \omega_0^2 + \frac{1}{2} \oint \rho \omega_0^2 A dz$ denotes the total energy stored by the moment of inertia of the unit and the inertia of the conveying liquid, $E_f$ refers to the energy of various losses in the coast-down transition, $\rho$ represents the density of the conveying liquid with units in kg/m$^3$, $A$ denotes the average sectional area of the loop pipe with the unit of m$^2$, $z$ refers to the effective pipeline length for the whole circuit with the unit of m, $t$ represents the time since the outage began with the unit of s, and $I$ refers to the total moment of inertia of the unit. With the unit of kg m$^2$, $P_0$, $\eta_0$, and $n_0$ are the effective power, efficiency, and rated speed of the nuclear reactor coolant pump under rated conditions, respectively, $n(t)$ refers to the rotational speed at different points in the coast-down transition with the unit of r/min.

Figure 1 shows the three-dimensional fluid calculation domain of the reactor coolant pump. In Figure 2, $\gamma$ denotes the outlet inclination of the impeller, $\beta_2$ denotes the outlet angle of the impeller, $\varphi$ denotes the wrap angle of the impeller, $Z$ denotes the blade numbers of the impeller, $D_2$ denotes the outlet diameter of the impeller in mm, $b_2$ denotes the outlet width of the impeller in mm, and $D_j$ denotes the inlet diameter in mm. Area ratio Y represents the ratio of the impeller outlet area to the volute throat area.

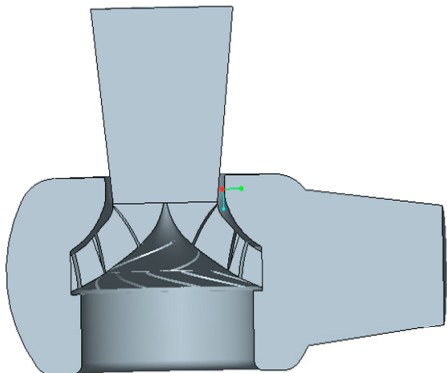

**Figure 1.** Three-dimensional fluid calculation domain of reactor coolant pump.

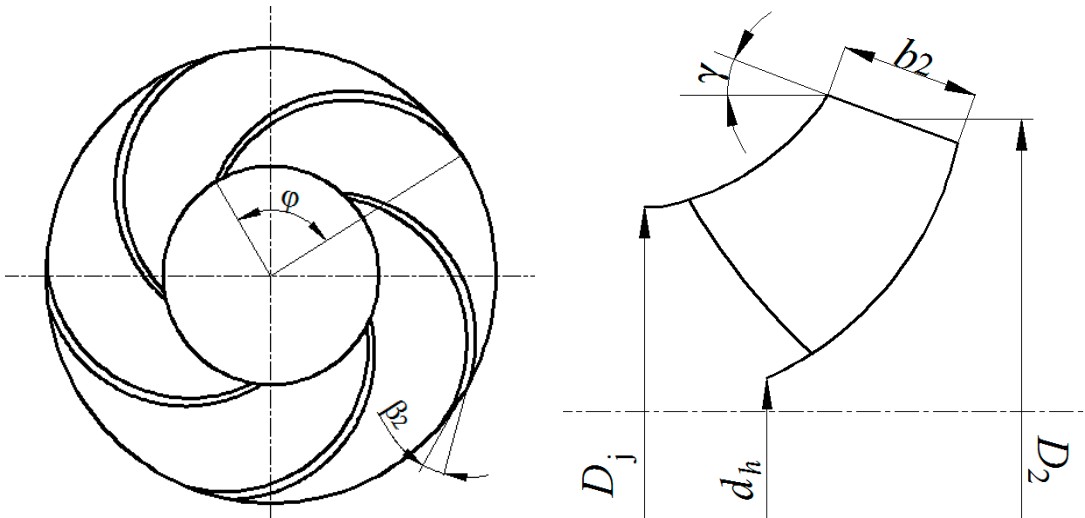

**Figure 2.** Schematic diagram of the main structural parameters of the impeller.

Table 1 presents that the efficiencies and heads obtained by the different combinations of eight different impeller geometric parameters, which suggest the influence of the different geometric parameters, parameter combination on efficiency, and the difference in the head [21]. Multiple linear regression was selected by this paper to analyze the relationship among each parameter, efficiency, and the head.

**Table 1.** Test scheme and performance calculation results.

|  | $\gamma/°$ | $\beta_2/°$ | $\varphi/°$ | $Z$ | $D_2$/mm | $b_2$/mm | $D_j$/mm | Area Ratio $Y$ | Index | |
|---|---|---|---|---|---|---|---|---|---|---|
|  |  |  |  |  |  |  |  |  | $\eta_1$ | $H$ |
| 1 | 20 | 20 | 115 | 4 | 760 | 190 | 555 | 0.927 | 79.45 | 94.89 |
| 2 | 20 | 25 | 120 | 5 | 765 | 190 | 560 | 0.916 | 81.09 | 109.48 |
| 3 | 20 | 30 | 125 | 6 | 770 | 195 | 550 | 0.916 | 82.76 | 134.33 |
| 4 | 23 | 30 | 125 | 5 | 765 | 195 | 550 | 0.932 | 84.38 | 111.69 |
| 5 | 23 | 25 | 120 | 6 | 760 | 200 | 560 | 1.002 | 84.70 | 138.02 |
| 6 | 23 | 20 | 115 | 4 | 760 | 200 | 555 | 0.952 | 84.04 | 100.39 |
| 7 | 26 | 20 | 115 | 4 | 770 | 190 | 555 | 0.914 | 82.24 | 101.80 |
| 8 | 26 | 25 | 120 | 5 | 765 | 190 | 560 | 0.925 | 82.16 | 112.89 |
| 9 | 26 | 30 | 120 | 6 | 765 | 195 | 550 | 0.935 | 82.32 | 136.20 |
| 10 | 20 | 20 | 125 | 6 | 770 | 195 | 550 | 0.916 | 80.67 | 101.59 |
| 11 | 23 | 30 | 125 | 4 | 770 | 200 | 560 | 0.931 | 83.10 | 95.47 |
| 12 | 26 | 25 | 115 | 5 | 760 | 200 | 555 | 0.956 | 82.90 | 119.49 |
| 13 | 20 | 30 | 115 | 6 | 770 | 190 | 555 | 0.905 | 84.22 | 137.83 |
| 14 | 23 | 30 | 120 | 5 | 765 | 190 | 560 | 0.921 | 83.69 | 124.90 |
| 15 | 26 | 25 | 120 | 6 | 760 | 195 | 550 | 0.946 | 82.27 | 120.81 |
| 16 | 20 | 20 | 115 | 4 | 770 | 195 | 550 | 0.916 | 83.98 | 101.89 |
| 17 | 23 | 20 | 125 | 5 | 765 | 200 | 560 | 0.941 | 83.11 | 100.60 |
| 18 | 26 | 25 | 125 | 4 | 760 | 200 | 555 | 0.956 | 81.31 | 103.49 |

*2.1. Data Normalization*

Data normalization was an important step in multiple linear regression. Given that each variable was different in the physical properties, they usually have different orders of magnitude and dimensions. When the regression equation was established directly, the regression coefficient was very often not directly comparable [21]. When variables vary largely in level, the role of the value of the higher numerical variable in the comprehensive analysis shall be highlighted, and the role of the comparatively lower numerical variable will be weakened if the raw data is directly used for analysis. Accordingly, to

ensure the reliability of the results, it was imperative for standardizing the table and simulation data to eliminate the dimensional influence between variables so that the data could be comparable. Data normalization involved the centralization and compression processing of the data simultaneously, the specific principles were as follows

$$x_{ij}^* = \frac{x_{ij} - \bar{x}}{s_j}, \qquad \begin{array}{l} i = 1, 2, 3, \ldots, n \\ j = 1, 2, 3, \ldots, p \end{array} \tag{3}$$

where, $x_{ij}$ represents the value of the $i$-th row and the $j$-th column, $x_{ij}$ * represents the normalized data of $x_{ij}$, $i$ refers to the $i$-th row, and $j$ refers to the $j$-th column, $s_j$ represents the normalized parameter of column $j$.

## 2.2. Path Analysis

Path analysis was conducted to analyze the direct relationship between the impeller geometrical parameters and the pump performance, as well as the indirect coupling relationship between the parameters. Assuming that the $p$ independent variable can be established, $x_1, x_2, \ldots, x_p$, the simple correlation coefficients between each of the two variables and the dependent variable $y$ were capable of forming the normalized normal equation to solve the path coefficient:

$$\begin{array}{l} r_{11}\rho_1 + r_{12}\rho_2 + \ldots + r_{1p}\rho_p = r_{1y} \\ r_{21}\rho_1 + r_{22}\rho_2 + \ldots + r_{2p}\rho_p = r_{2y} \\ \quad \ldots \quad \ldots \quad \ldots \quad \ldots \\ r_{p1}\rho_1 + r_{p2}\rho_2 + \ldots + r_{pp}\rho_p = r_{py} \end{array} \tag{4}$$

where, $\rho_1, \rho_2, \ldots, \rho_p$ was the direct path coefficient. The direct path coefficient represented the direct effect size of the independent variable, while the indirect path coefficient suggested that the independent variable influences the dependent variable by impacting other independent variables. Besides, such coefficients could be calculated using the correlation coefficient $r_{ij}$ and the direct path coefficient $\rho_i$. The direct path coefficient can be obtained by calculating the inverse matrix of the noted correlation matrix. With the assumption that $B_{ij}$ was the inverse matrix of the correlation matrix $r_{ij}$, and then the direct path coefficient $\rho_i$ ($i = 1, 2, \ldots, p$) was expressed as follows

$$\begin{bmatrix} \rho_1 \\ \rho_2 \\ \ldots \\ \rho_p \end{bmatrix} = \begin{bmatrix} B_{11} & B_{12} & B_{13} & \ldots & B_{1p} \\ B_{21} & B_{22} & B_{23} & \ldots & B_{2p} \\ \ldots & \ldots & \ldots & \ldots & \ldots \\ B_{p1} & B_{p2} & B_{p3} & \ldots & B_{pp} \end{bmatrix} \begin{bmatrix} r_{1y} \\ r_{2y} \\ \ldots \\ r_{py} \end{bmatrix} \tag{5}$$

The path coefficient $\rho_{ye}$ of the remaining term is as expressed in Equation (6). In case the path coefficient $\rho_{ye}$ of the remaining term was smaller, the impeller geometrical parameters and the performance would be well satisfied with the linear relation. Conversely, when the path coefficient $\rho_{ye}$ of the remaining term was larger, it suggested that the test error was larger or other important factors were not introduced.

$$\rho_{ye} = \sqrt{1 - \left(\sum_{i=1}^{p} r_{iy}\rho_i\right)} \tag{6}$$

By path analyzing the impeller geometrical parameters and the pump head and the efficiency performance, the results were as listed in Table 2.

**Table 2.** Path analysis results between impeller geometric parameters and performance.

| Factor | Direct Effect | Indirect Effect | | | | | | | |
|---|---|---|---|---|---|---|---|---|---|
| | | $\gamma \to H$ | $\beta_2 \to H$ | $\varphi \to H$ | $Z \to H$ | $D_2 \to H$ | $b_2 \to H$ | $D_0 \to H$ | $Y \to H$ |
| $\gamma$ | 0.0076 | | 0.0424 | 0.0000 | −0.0502 | −0.0748 | −0.0365 | −0.0031 | 0.1825 |
| $\beta_2$ | 0.5091 | 0.0006 | | −0.1244 | 0.2510 | 0.0299 | 0.0122 | 0.0000 | −0.0349 |
| $\varphi$ | −0.3731 | 0.0000 | 0.1697 | | 0.1506 | 0.0299 | −0.0487 | 0.0000 | 0.0295 |
| $Z$ | 0.6025 | −0.0006 | 0.2121 | −0.1041 | | 0.015 | 0.0122 | 0.0092 | 0.0322 |
| $D_2$ | 0.1795 | −0.0032 | 0.0849 | −0.0622 | 0.0502 | | 0.0487 | 0.0061 | −0.3235 |
| $b_2$ | −0.1461 | 0.0019 | −0.0424 | −0.1244 | −0.0502 | −0.0598 | | 0.0000 | 0.3087 |
| $D_0$ | −0.0368 | 0.0006 | 0.0000 | 0.0000 | −0.1506 | −0.0299 | 0.0000 | | 0.1007 |
| $Y$ | 0.4401 | 0.0032 | −0.0404 | −0.0250 | 0.0441 | −0.1320 | −0.1025 | −0.0084 | |

| Factor | Direct Effect | Indirect Effect | | | | | | | |
|---|---|---|---|---|---|---|---|---|---|
| | | $\gamma \to \eta_1$ | $\beta_2 \to \eta_1$ | $\varphi \to \eta_1$ | $Z \to \eta_1$ | $D_2 \to \eta_1$ | $b_2 \to \eta_1$ | $D_0 \to \eta_1$ | $Y \to \eta_1$ |
| $\gamma$ | −0.0528 | | 0.0400 | 0.0000 | 0.0001 | −0.2726 | 0.1070 | 0.0040 | 0.2509 |
| $\beta_2$ | 0.4803 | −0.0044 | | −0.1636 | −0.0005 | 0.1090 | −0.0269 | 0.0000 | −0.048 |
| $\varphi$ | −0.4908 | 0.0000 | 0.1601 | | −0.0003 | 0.1090 | 0.1076 | 0.0000 | 0.0406 |
| $Z$ | −0.0011 | 0.0044 | 0.2001 | −0.1227 | | 0.0545 | −0.0269 | −0.0120 | 0.0443 |
| $D_2$ | 0.6542 | 0.022 | 0.0800 | −0.0818 | −0.0001 | | −0.1076 | −0.0080 | −0.4445 |
| $b_2$ | 0.3228 | −0.0132 | −0.0400 | −0.1636 | 0.0001 | −0.2181 | | 0.0000 | 0.4242 |
| $D_0$ | 0.0480 | −0.0044 | 0.000 | 0.0000 | 0.0003 | −0.1090 | 0.0000 | | 0.1383 |
| $Y$ | 0.6047 | −0.0219 | −0.0381 | −0.0329 | −0.0001 | −0.4809 | 0.2265 | 0.0110 | |

## 3. Results

### 3.1. The Direct Impact Analysis of the Main Geometric Parameters of Nuclear Reactor Coolant Pump on Its Performance

As is evident from Table 2 the blade number, blade outlet angle, blade wrap angle, area ratio, impeller outlet diameter, and the blade outlet width in the eight impeller geometrical parameters had a large direct impact on the design point head of the nuclear reactor coolant pump, and the other two parameters had very little impact on the head. Among the six parameters with larger direct impact, the blade numbers were the largest; the blade outlet angle, blade wrap angle, area ratio, and impeller outlet diameter were ranked second; and the blade outlet width was the smallest. The direct path coefficient of the impeller blade number was 0.6025, suggesting that the blade number was the most critical in the geometrical parameters of the reactor coolant pump. When blade numbers were changed, the work efficiency of the impeller changed greatly, and its head also changed significantly. In a particular range, the head of the pump would definitely rise with the increase in the blade numbers. The direct path coefficients of the impeller blade outlet angle and area ratio were 0.5091 and 0.4401, respectively, which confirmed that the blade outlet angle and the area ratio on the head of the nuclear reactor coolant pump also had a major role. When the blade outlet angle was changed, the circumferential component of the absolute velocity at the impeller outlet was also changed. Thereafter, the circumferential component of the absolute velocity at the impeller outlet increased, while the head of the pump increased with the increase of blade outlet angle. As for area ratio the reaction of impeller and guide vane matching relationship, one of the important physical quantities, the performance of the pump was not unilaterally decided by the impeller but by the impeller, guide blade, and volute both (in this study, the volute remains the same, so it need not be considered). In the design process, the inlet area of the guide vane had hardly changed; hence the increase of area ratio in a certain scope was equivalent to the reduction in the impact loss and made the head increase. For the change of the blade wrap angle, the binding force of the fluid in the impeller channel was changed, and the relative velocity liquid angle of the impeller outlet was also changed. In a specific range, the restraint of the fluid in the impeller channel increased, while the blade wrap angle increased, however, the relative flow angle of the impeller outlet decreased, with the head. The direct path coefficient of the impeller outlet diameter was 0.1795, which indicated that in a certain range, the energy of the fluid increases, and the head also rises. The direct path coefficient of impeller outlet width minimum was −0.1461, which suggested that within a certain

range, an increase of blade outlet width can make the impeller outlet edge overtilted, and a larger secondary flow would appear, resulting in the head decrease.

From Table 1, it can be observed that amongst the eight impeller geometric parameters, the blade outlet angle, blade wrap angle, blade outlet width, impeller outlet diameter, and area ratio had a greater direct impact on the efficiency performance of nuclear reactor coolant pump design point, while the other three parameters had a relatively smaller impact on its efficiency. Among the five parameters with larger direct impact, the impeller outlet diameter was the largest; the blade outlet angle, blade wrap angle, and area ratio ranked second; and blade outlet width was the smallest. The direct path coefficient of impeller outlet diameter reached 0.6542, which confirmed that impeller outlet diameter was the most critical of all the geometrical parameters of the nuclear reactor coolant pump.

The efficiency of nuclear reactor coolant pump varies with the impeller outlet diameter and the flow condition of the impeller outlet. This indicated that, within a certain range, with the increase of the impeller outlet diameter the impeller outlet speed is reduced, the impact loss between the blade wheel and guide vane decreases, and the hydraulic efficiency of the pump increases. The direct path coefficient of the area ratio between the impeller and the guide vane was 0.6047, which substantiated that increasing the impeller outlet area in a certain range makes the impeller and guide vane match better, reducing the impact loss and increasing the efficiency. The direct path coefficient of the impeller blade outlet angle and the blade wrap angle were 0.4803 and −0.4908, respectively, which confirmed that the blade angle and the blade wrap angle exert the main impact on the design point efficiency of the nuclear reactor coolant pump. Additionally, within a certain range, with the increase in the blade outlet angle, the circumferential component of the absolute speed at the impeller outlet increases, and the efficiency also increases. With the increase of the blade wrap angle, the fluid in the flow channel [22] gets restrained by the stronger blades, while the excessive flow channel increases the friction loss and decreases the efficiency of the pump. The direct path coefficient of the blade outlet width reached 0.3228, which suggested that in a particular range, with the increase of the blade outlet width, the pump can increase the efficiency.

*3.2. Indirect Effect Analysis of the Geometric Parameters of the Nuclear Reactor Coolant Pump on Its Performance*

Besides the direct impact on pump performance, the impeller geometry parameters have different degrees of mutual influence. The principle of path analysis is Correlation coefficient = direct path coefficient + indirect path coefficient, i.e., when a parameter changes, it not only has a direct impact on the performance, but also exerts an indirect impact on the performance by changing the other geometric parameters.

From the indirect path coefficient listed in Table 2, it is evident that for the head index, the indirect path coefficient of the outlet lean angle and the impeller inlet diameter were 0.0603 and −0.0792, respectively. This indicates that the outlet lean angle and the impeller inlet diameter exerted little indirect impact on the head by changing the other geometrical parameters, primarily by changing the area ratio ($\gamma \rightarrow Y \rightarrow H = 0.1825$, $D_0 \rightarrow Y \rightarrow H = 0.1007$). The indirect path coefficient of the blade outlet angle was 0.1344, indicating that blade outlet angle indirectly strengthens the head by changing the other geometrical parameters, and the outlet lean angle had the effect of reducing the head by changing the blade wrap angle ($\beta_2 \rightarrow \varphi \rightarrow H = -0.1244$). However, the outlet lean angle reinforced the head by blade numbers ($\beta_2 \rightarrow Z \rightarrow H = 0.251$). The indirect path coefficient of the blade wrap angle was 0.311, indicating that the blade wrap angle had an indirect strengthening effect on the head by changing the other geometrical parameters, and the blade wrap angle strengthens the head by changing the blade outlet angle and the blade numbers ($\varphi \rightarrow \beta_2 \rightarrow H = 0.1697$, $\varphi \rightarrow Z \rightarrow H = 0.1506$). The indirect path coefficient of the impeller blade numbers was 0.1868, suggesting that the impeller blade numbers exerted little indirect impact on the head by changing the other geometrical parameters, and the impeller blade numbers strengthened the head by changing the blade outlet angle ($Z \rightarrow \beta_2 \rightarrow H = 0.2121$), and yet the impeller blade numbers reduced the head by the blade wrap angle ($Z \rightarrow \varphi \rightarrow H$

= −0.1041). The indirect path coefficient of the blade outlet width was 0.0338, indicating that the blade outlet width had slightly enhanced the indirect impact on the head by changing the other geometrical parameters, and the blade outlet width reduced the head by changing blade wrap angle ($b_2 \rightarrow \varphi \rightarrow H$ = −0.1244), and yet the blade outlet width had the effect of reinforcing the head by area ratio ($b_2 \rightarrow Y \rightarrow H$ = 0.3087). The indirect path coefficient of the impeller outlet diameter was −0.199, which implied that the impeller outlet diameter had an abridged indirect effect on the head by changing the other geometrical parameters, and the impeller outlet diameter reduced the head by changing area ratio ($D_2 \rightarrow Y \rightarrow H$ = −0.3235). The indirect path coefficient of the area ratio was −0.261, which suggests that the area ratio had an abridged effect on the head, and the area ratio reduced the head by changing the impeller outlet diameter and the blade outlet width ($Y \rightarrow D_2 \rightarrow H$ = −0.132, $Y \rightarrow b_2 \rightarrow H$ = −0.3235).

For the efficiency index, the indirect path coefficient of impeller outlet diameter was 0.0252, which proved that the impeller inlet diameter had a small indirect impact on the efficiency by changing the other geometrical parameters. The impeller inlet diameter impacted the efficiency by the impeller outlet diameter and the area ratio, while the impeller inlet diameter reduced the efficiency by changing the impeller outlet diameter ($D_0 \rightarrow D_2 \rightarrow \eta_1$ = −0.109), the impeller inlet diameter increased the efficiency by the area ratio ($D_0 \rightarrow Y \rightarrow \eta_1$ = 0.1383). The indirect path coefficient of the outlet lean angle was 0.1031, which indicated that the outlet lean angle had little indirect impact on the increase of the efficiency by changing the other geometrical parameters, the outlet lean angle increased the efficiency by the blade outlet width and the area ratio ($\gamma \rightarrow b_2 \rightarrow \eta_1$ = 0.107, $\gamma \rightarrow Y \rightarrow \eta_1$ = 0.2509), while it decreased the efficiency by area ratio ($\gamma \rightarrow D_2 \rightarrow \eta_1$ = −0.2726). The indirect path coefficient of the blade outlet angle was −0.1344, showing that blade outlet angle had an indirect impact on the decrease of the efficiency by changing the other geometrical parameters, and the blade outlet angle decreased the efficiency by the blade wrap angle ($\beta_2 \rightarrow \varphi \rightarrow \eta_1$ = −0.1636); the blade outlet angle decreased the efficiency by the impeller outlet diameter ($\beta_2 \rightarrow D_2 \rightarrow \eta_1$ = 0.109). The indirect path coefficient of the blade wrap angle was −0.417, hinting that the blade wrap angle had an indirect impact on the increase of the efficiency by changing the other geometrical parameters, and the blade wrap angle increased the efficiency by blade outlet angle, impeller outlet diameter, and the blade outlet width ($\varphi \rightarrow \beta_2 \rightarrow \eta_1$ = 0.1601, $\varphi \rightarrow b_2 \rightarrow \eta_1$ = 0.109, $\varphi \rightarrow b_2 \rightarrow \eta_1$ = 0.1076). The indirect path coefficient of the impeller blade numbers was 0.1417, which demonstrated that the impeller blade numbers had an indirect impact on the increase of the efficiency by changing the other geometrical parameters and the impeller blade numbers increased the efficiency by the blade outlet angle ($Z \rightarrow \beta_2 \rightarrow \eta_1$ = 0.2001), while the impeller blade numbers decreased the efficiency by the blade wrap angle ($Z \rightarrow \varphi \rightarrow \eta_1$ = −0.1227). The indirect path coefficient of the impeller outlet diameter was −0.54, denoting that the impeller outlet diameter had an indirect impact on the decrease of the efficiency by changing the other geometrical parameters, and then it decreased the efficiency by the blade outlet width and the area ratio ($D_2 \rightarrow b_2 \rightarrow \eta_1$ = −0.1076, $D_2 \rightarrow Y \rightarrow \eta_1$ = −0.4445). The indirect path coefficient of the blade outlet width was −0.0106, which implied that the blade outlet width had an indirect impact on the decrease of the efficiency by changing the other geometrical parameters, and the blade outlet width decreased the efficiency by the area ratio ($b_2 \rightarrow Y \rightarrow \eta_1$ = 0.4242), while it decreased the efficiency by the blade wrap angle and the impeller outlet diameter ($b_2 \rightarrow \varphi \rightarrow \eta_1$ = −0.1636, $b_2 \rightarrow D_2 \rightarrow \eta_1$ = −0.2181).

It was acquired by analyzing the indirect path coefficient between the different geometric parameters that the influence weight of each parameter was different when the different performances served as the index ($\beta_2 \rightarrow Z \rightarrow H$ = 0.251, $\beta_2 \rightarrow Z \rightarrow \eta_1$ = −0.0005). Under the index of the same performance, the influence weight of each parameter was directional ($\beta_2 \rightarrow \varphi \rightarrow H$ = −0.1244, $\varphi \rightarrow \beta_2 \rightarrow H$ = 0.1697). Under the small indirect path coefficient, it did not mean that there was little interaction between the factor and other factors (the indirect path coefficient of the impeller outlet diameter reaches −0.0252, $D_0 \rightarrow D_2 \rightarrow \eta_1$ = −0.109, $D_0 \rightarrow Y \rightarrow \eta_1$ = 0.1383), which neutralized the indirect effect on each parameter.

### 3.3. Analysis of Residual Path Coefficient

The residual path coefficient is a critical value that determines the satisfaction of the linear relation between parameters and performances. In this paper, the determined path coefficients and the residual path coefficients between the eight geometrical parameters of the impeller and the performance of the nuclear reactor coolant pump are listed in Table 3.

**Table 3.** Residual path coefficient between impeller geometric parameters and performance.

| Performance | Determine Path Coefficient | Residual Path Coefficient |
|:---:|:---:|:---:|
| $H$ | 0.8421 | 0.2909 |
| $\eta_1$ | 0.6678 | 0.5541 |

It can be seen from the table that the determined path coefficient between the head and the performance reached 0.8421, which indicated that the eight parameters selected could calculate it accurately through the linear relationship, while the path coefficient of efficiency was determined as 0.6678. This proved that selecting the eight parameters and efficiency cannot satisfy the linear relationship, there may be larger errors, or the main parameter was not selected. However, according to the actual situation, as the flow of the components of nuclear reactor coolant pump, the impeller was employed to convert mechanical energy into potential energy, so that the head primarily becomes dependent on the impeller geometric parameters. For efficiency, the impeller was only a part of the nuclear main pump flow components, and the influence of the impeller geometry on the efficiency was greater than the selected seven. Thus, the determined path coefficient was normally not large, and hence the calculation process was accurate. The eight parameters selected could have been used as a representation of efficiency by linear. To visually represent the relationship between the efficiency index, the lift index, and the parameters, please refer the path diagram (shown in Figure 3).

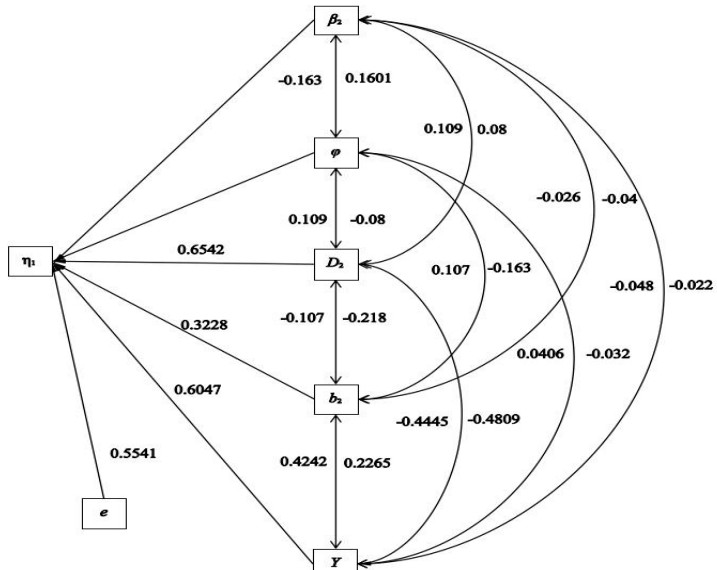

(**a**) The path diagram between the impeller geometrical parameters and the efficiency.

**Figure 3.** *Cont.*

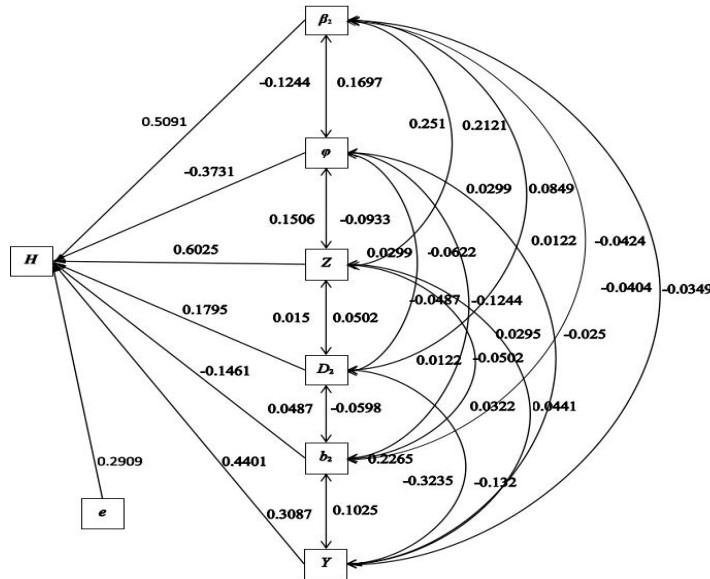

(**b**) The path diagram between the impeller geometrical parameters and the head

**Figure 3.** The path diagram between the impeller geometrical parameters and the performance.

### 3.4. Optimal Parameter Selection

By selecting the eight major parameters of the impeller, efficiency, and the head, respectively, as the performance evaluation indices, the geometrical parameters of the different indices have proved to have different effect sizes. When efficiency was the performance index, five geometrical parameters exerted the greatest influence on efficiency in line with the size of influence weight: blade outlet angle > impeller outlet diameter > blade wrap angle > area ratio > blade outlet width. When the head was the performance index, the six geometric parameters had the greatest impact on efficiency; according to influence weight: impeller blade numbers > blade outlet angle > blade wrap angle > area ratio > impeller outlet diameter > blade outlet width. This paper considers efficiency as the main performance index, the optimal parameters were selected in line with the results of partial correlation analysis and path analysis, and the results were as listed under Table 4.

**Table 4.** Optimal combination of impeller geometrical parameters.

| Factor | $\gamma$ | $\beta_2$ | $\varphi$ | Z | $D_2$ | $b_2$ | $D_0$ | Y |
|---|---|---|---|---|---|---|---|---|
| Optimal results | 23° | 30° | 115° | 5 | 770 mm | 200 mm | 555 mm | 1.002 |

## 4. Experiment Verification

To verify the effectiveness of this optimization method, the model pump, developed as per the specified parameters, was tested and verified. The model pump had the following specifications; design flow $Q_M$ = 104 m$^3$/h, head $H_M$ = 3.6 m, rotating speed $n$ = 1480 r/min, and specific speed $n_s$ = 351. The model pump and the reactor coolant pump size had a ratio of 5.56. The transient performance testbed for reactor coolant pump was presented in Figure 4. To complete the collection, the flow measurement used the LWGY-type turbine flow sensor instrument supplied by the Nanjing Ditai Electromechanical Equipment Co. Ltd (Nanjing, China), turbine flow meter diameter of DN125, output current signal of 4–20 mA, and acquisition accuracy of 0.5 grade. The instantaneous flow signal was acquired from Beijing Altai Technology Development Co., Ltd. We used a production USB3200 type data acquisition card with rotating speed function and a moment sensor: ZJ-type rotating speed and the moment sensor supporting WJCG dynamometer acquisition pump shaft speed, moment, and other data; its working principle was magnetoelectric conversion and electric phase difference, the

measurement range of the moment was 0–50 NM, the number of teeth was 180, the precision was 0.2%, and the speed range was 0–5000 r/min. In the test process, the model pump was first, and then the outlet valve was adjusted after its operations were stabilized so that the pump could be shut down under the rated working condition. The start-up curves of the three groups with starting time of approximately 2 s, 4.5 s, and 8.5 s, respectively, were obtained through the coupling to connect the flywheels with different moments of inertia to reduce the starting acceleration of the unit. Through the similar conversion of the model pump, the starting characteristic curves of the three groups of different starting accelerations of the nuclear reactor coolant pump were determined as in Figure 5.

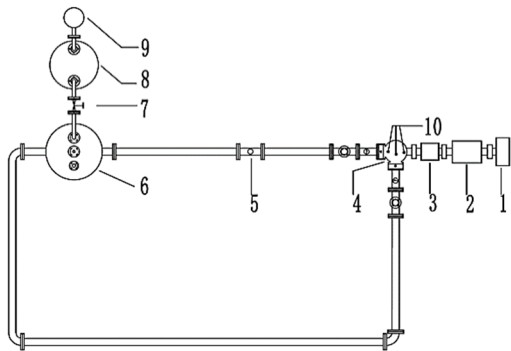

**Figure 4.** Test device for the coast-down transition process of the nuclear reactor coolant pump.

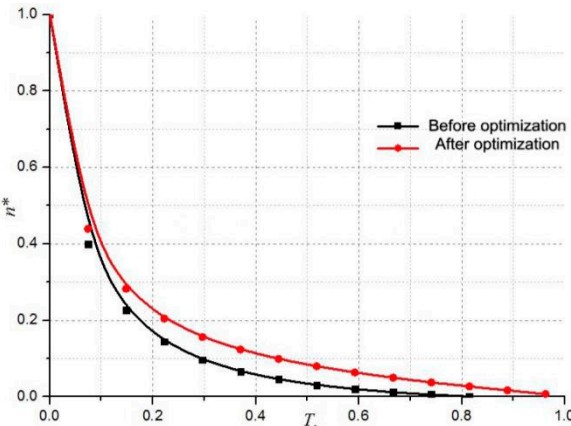

**Figure 5.** Change of rotational speed before and after the optimization of the nuclear reactor coolant pump during the coast-down transition process.

A dimensionless conversion was carried out with the optimized coast-down time of the pump as a cycle. From Figure 5, it is evident that the optimization had a great influence on the change of the rotation speed [23] in the coast-down transition of the reactor coolant pump. Because of the small moment of inertia of the impeller itself, the small changes in the impeller geometrical parameters and the amount of change in the moment of inertia can be neglected. Figure 3 illustrates that the curves of speed variations in the coast-down transition were different before and after the optimization. The preoptimization speed decreased to zero at nearly $0.8\ T_1$, and the optimized speed dropped to zero at nearly $1\ T_1$. Throughout the coast-down transition, the size of the speed before optimization was greater than the size of the speed of coast-down after optimization. This proved that almost the entire coast-down characteristics had been improved through the structural optimization of the nuclear reactor coolant pump.

### 5. Conclusions

The inertia moment energy stored in the rotor system was utilized to keep the nuclear reactor coolant pump running during coast-down transition. Considering the additional losses of the impeller were primarily caused by the nuclear reactor coolant pump under the off-design condition, the energy loss was reduced and the time of coast-down was delayed by optimizing the main structural parameters of the impeller.

(1) According to the energy conservation law, the calculation equation of coast-down time and the energy utilization of the inertia moment storage were listed, and the basis of coast-down optimization was given based on the reducing energy loss in the coast-down transition. Hydraulic optimization design of the reactor coolant pump impeller was carried out by combining orthogonal optimization test and CFD simulation software, while the hydraulic characteristics of the calculation model were also completed.

(2) The correlation between the impeller geometric parameters and efficiency, head, and different geometric parameters was computed, and the main parameters affecting efficiency and the pressure head were determined. By the path analysis on the results of hydraulic characteristics, the direct influence of geometric parameters on efficiency and head and the indirect influence of the geometric parameters on other parameters that changed the efficiency and head were ascertained.

(3) The efficiency of the pump was the target, the head was the constraint condition, and, combined with the calculation results of the partial correlation analysis and path analysis, the optimal parameters were selected as $\gamma = 23^{\circ}$, $\beta_2 = 30^{\circ}$, $\varphi = 115^{\circ}$, $Z = 5$, $b_2 = 200$ mm, $D_2 = 770$ mm, $D_0 = 555$ mm, and $Y = 1.002$, respectively.

**Author Contributions:** Y.Z.: Experiments and simulations; X.S.: The writing and revision of the paper; X.W.: Ideas and fund support of the paper; R.Z., Q.F. and H.Z.: Experimental and simulated data processing.

**Funding:** The National Natural Science Foundation of China (51379091); a science and technology program funded by the Natural Science Foundation of Jiangsu Province (BK20130516); the National Youth Natural Science Foundation of China (51509112); key R & D programs of Jiangsu Province of China (BE2015129 & BE2016160); and a prospective joint research project of Jiangsu Province (BY2016072-02).

**Conflicts of Interest:** The authors declare no conflict of interest.

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
