# Peer review of "The Influence and Optimization of Geometrical Parameters on Coast-Down Characteristics of Nuclear Reactor Coolant Pumps"

_processes, doi:10.3390/pr7060327_

Round 1
Reviewer 1 Report
The Influence and Optimization of Geometrical Parameters on Coast-down Characteristics of Nuclear Reactor Coolant Pump
General comments
The paper describe a method for optimize the design of the coolant pump using orthogonal method.
English is very poor and need serious revision by a native English speaker. I think that at least one scheme of the cooling system and a picture of the coolant pump is required. A graphical comparison between a classical pump and the optimized one have to be presented. As the paper lack of a comprensible Language, I stop my revison at line 160. Was impossible for me in continue the revision due to the bad english used. Please make a global english editing and then I will happy to revise it.Detailed comments.
Abstract
English is poor and need to be revised. Line 15-18, one sentence, is unreadable. The statement “
avoid serious nuclear accidents“: I think that the coolant pump is not the sole cause of nuclear accidents.
Text
Line 22-25: sentence not clear.
Line 26-27: what is the cooling circuit working time?
Line 28: “cause hydrogen concentration” why?
Line 32. “At home” of the authors?
Line 33 and following: the list have to separated by, for example, by a numbered list: (i) first statement; (ii) second..; (iii) third.
Line 42: TRR – not defined
Line 46-48: too general. If there are few researches about the factor that affect the coast-down characteristics, please cite them.
Line 50-52: I think that is not needed to have 3D numerical simulation to discover that the geometry of the impeller is affecting the efficiency of a pump. May be you have to put focus your attention on some particular parameters.
Line 57: “The noted literature…” too general. Please specify.
Line 60-62: not clear, please reformulate.
Line 72: Can you add reference to this?
Line 76: what mean maximize improve?
Line 76-79:
Line 79-80: can you give references to these high requirements?
Line 80-81: otherwise of what?
Line 89-92: Not clear. The sentence is not correct.
Eq. 1 and Eq 2 are not explained. Please specify the meaning of each variables of eq.1, 2.
Eq. 95: usually density is given in kg/m3. Here there is need to give this information.
Tab 1. All the variables of the table was not explained in the text or in some picture.
Line 104-105. Before resenting results, authors must introduce something about the number solutions investigated, why 8, what kind of combination, etc.
Eq. 3,4,5: not all symbols are explained
Also, please give reference to this method of data normalization.
Eq. 6 not all symbols are explained.
Line 134: one of them? I don’t understand
Line 134-140: what is the correlation matrix? Any reference to it?
Line 141: not clear
Line 141: is smaller of what?
Tab 2: similar comments as tab 1.
Line 150: very difficult to understand
Line 155-160 too long sentence, please reformulate.
Author Response
Line 22-25: sentence not clear.
Reply: These sentences have been revised. It has been modified to “Nuclear reactor coolant pump is the key equipment in the primary loop cooling system of nuclear power plant. Once a power failure occurs in a nuclear power plant, the reactor coolant pump loses its power source, and it will enter the coast-down state. In a short time, the inertia of the inert wheel provides power for the reactor coolant pump, and the coolant can continue to cool the reactor core. “
Line 26-27: what is the cooling circuit working time?
Reply: Because the coolant pump loses energy power, it only relies on the kinetic energy stored in the inert wheel. When the kinetic energy is exhausted, the impeller stops working on the coolant. This time is about 6-7 seconds.
Line 28: “cause hydrogen concentration” why?
Reply: These sentences have been revised. It has been modified to “The heat of the reactor can not be discharged in time and the temperature rises sharply, which can easily cause the decomposition of coolant and generate a large amount of hydrogen, which is not conducive to the safety of the system.”
Line 32. “At home” of the authors?
Reply: “at home and abroad” has been modified to “domestic and foreign”
Line 33 and following: the list have to separated by, for example, by a numbered list: (i) first statement; (ii) second..; (iii) third.
Reply: Delete the sentence " and the main examples are as follows:" here.
Line 42: TRR – not defined
Reply: “the main TRR circuit” has been modified to “primary circuit cooling system”
Line 46-48: too general. If there are few researches about the factor that affect the coast-down characteristics, please cite them.
Reply: some references about the factor that affect the coast-down characteristics were cited in the following description.
Line 50-52: I think that is not needed to have 3D numerical simulation to discover that the geometry of the impeller is affecting the efficiency of a pump. May be you have to put focus your attention on some particular parameters.
Reply: Numerical simulation can reduce the cost of research, and numerical simulation has great reference value. In the previous research, the method of prolonging coast-down time can be analyzed by numerical calculation.
Line 57: “The noted literature…” too general. Please specify.
Reply: it should be “The noted literature mentioned above “.
Line 60-62: not clear, please reformulate.
Reply: It has been modified to “For the influencing factors of coast-down characteristics of reactor coolant pump, the indirect effects of coupling effects between different geometric parameters combinations are studied in depth in this paper.”
Line 72: Can you add reference to this?
Reply: The corresponding literature is marked in Line 72.
Line 76: what mean maximize improve?
Line 76-79:
Reply: Because the greater the moment of inertia, the more energy stored in the rotor system, the longer the idling time will be when power failure occurs. We expect to have a longer inertia time, so we need to achieve the maximum inertia within the allowable range.
Line 79-80: can you give references to these high requirements?
Line 80-81: otherwise of what?
Reply: This sentence is meaningless to this article and has been deleted. “There are high requirements of structure and machining accuracy to ensure that the flywheel can remain intact and nondestructive in extreme cases. Otherwise, the flywheel will have a large production cost.”
Line 89-92: Not clear. The sentence is not correct.
Reply: It has been modified to “As the rotational inertia of the impeller is obviously smaller than that of the flywheel, the moment of inertia of the rotor basically remains unchanged when the geometric parameters of the impeller change.”
Eq. 1 and Eq 2 are not explained. Please specify the meaning of each variables of eq.1, 2.
Reply: In eq.1, 2, specify the meaning of each variables have been explained “In the equation [1, 2], ET denotes the total energy stored by the moment of inertia of the unit and the inertia of the conveying liquid; Ef refers to the energy of various losses in the coast-down transition; ρ represents the density of conveying liquid, with the unit is g/m3; A denotes the average sectional area of the loop pipe, with the unit of m2; z refers to the effective pipeline length for the whole circuit, with the unit of m; t represents the time since the outage began, with the unit of s; I refers to the total moment of inertia of the unit. With the unit of kg•m2; P0, η0 and n0 are the effective power, efficiency and rated speed of the nuclear reactor coolant pump under rated conditions, respectively; n(t) refers to the rotational speed at different points in the coast-down transition, with the unit of r/min.”
Eq. 95: usually density is given in kg/m3. Here there is need to give this information.
Reply: it should be “kg/m3”
Tab 1. All the variables of the table was not explained in the text or in some picture.
Reply: The schematic diagram of the structural parameters was added.
Line 104-105. Before resenting results, authors must introduce something about the number solutions investigated, why 8, what kind of combination, etc.
Reply: Because the impeller design is mainly includes eight parameters: Outlet inclination, outlet angle, Wrap angle, Number of blade, Export diameter, Outlet width,Inlet diameter, Area ratio.
Eq. 3,4,5: not all symbols are explained
Reply: “Where, xij represents the corresponding number of sampling data; sj represents the normalized parameter of column j; x*ij represents the data after normalized.” have been added.
Also, please give reference to this method of data normalization.
1. Reply: The reference on data normalization have been added. “Wu S T, Hou F H, Dai F. Linear approximation method in the process of standardization of non-linear data [J]. Journal of Information Engineering University, 2007, 02:250-253.”
Eq. 6 not all symbols are explained.
Reply: All symbols in Eq. 6 have been explained in the following description.
Line 134: one of them? I don’t understand
Reply: “one of them” should be “where”, which for explain the symbols in Eq. 6.
Line 134-140: what is the correlation matrix? Any reference to it?
Line 141: not clear
Line 141: is smaller of what?
Reply: Line 134-141 is the sentence describing these symbols in Eq. 6 and 7 .
Tab 2: similar comments as tab 1.
Reply: The schematic diagram of the structural parameters was added.
Line 150: very difficult to understand
Line 155-160 too long sentence, please reformulate.
Reply: The description in lines 150-166 was modified.

Reviewer 2 Report
Main flaws to be addressed:
1) English language should be completely revised (I suggest a complete re-writing of the manuscript by a native English speaker). Most parts do undermine the readers' comprehension.
2) All the geometrical features shown in Table 1 should have a correspondance in a 3D model or at least with a sketch, for a better understanding of the problem.
3) The paper should be thoroughly re-organized in all its sections. The method is reported (Data normalization and path analysis) in the Results paragraph and should instead be mentioned earlier.
Author Response
1) English language should be completely revised (I suggest a complete re-writing of the manuscript by a native English speaker). Most parts do undermine the readers' comprehension.
Reply: Some English descriptions of the paper have been revised.
2) All the geometrical features shown in Table 1 should have a correspondance in a 3D model or at least with a sketch, for a better understanding of the problem.
Reply: The structure diagram of the nuclear main pump was added.
3) The paper should be thoroughly re-organized in all its sections. The method is reported (Data normalization and path analysis) in the Results paragraph and should instead be mentioned earlier.
Reply: Paragraphs of the paper were reorganized as required.

Round 2
Reviewer 1 Report
The Influence and Optimization of Geometrical Parameters on Coast-down Characteristics of Nuclear
Reactor Coolant Pump
The paper was partially improved but still need revision. A lot of sentences still remain a uncompressible English and need to be revised. Not all formula symbols used are explained.
For these reason I ask again for major revision.
Line 10-11: “ The coast-down transition is to long maintain the cooling loop system using the energy stored by the high moment of inertia of the reactor coolant pump unit.”. Sentence is not clear
Line 11-12: “Due to the moment of inertia, the unit requires a high reliability of the nuclear reactor coolant pump and consumes considerable energy…” not clear. Please reformulate.
Line 15-17: the cost down ability? I do not understand
Line 22: sentence too strong: “coolant pump is the key”. Reference to this statement?
Line 24: In a short time: seconds, minutes?
Line 50 of the impeller
Line 57 CFD?
Line 69: the geometric parameters, the efficiency and the hydraulic head.
Line 90-91: the equation 1 give Et, but at line 91 the equation of Et is different. It is right?
Fig 2 : caption: describe symbols used. Give separate explanation for the two scheme.
Equation 3: what are x*ij, xij, x, n, P?
Equation 4: what is Sj also?
I suggest to remove equation 3 and 4 and give only eq. 5.
Table 1. What is Y area ratio?
Table 2: as the coefficient are “adimesional”, they have same accuracy, so they have the same number of decimal digits
Line 161, 204 punctuation (or upper case) errors
Figure 5: measurements unit for T1?
Line 312-313: please reformulate in a clear sentence.
Line 313: “hydraulic head”
Line 318: acquisition accuracy of 0.5->missing measurement unit
Line 324 precision is 0.2, missing measurement unit.
Line 347-348: Against energy of unit moment of inertia is stored to keep nuclear reactor coolant pump running 347 in the coast-down transition->sentence is not clear and need to be rewritten in an appropriate English.
Line 348:Given that the additional losses of the->loses of what?
Line 354-356: sentence too long and not clear. Upper case/lower case and punctuation errors.
Line 357: what are the external characteristic results? Also they are not “calculation models”, may be they are numerical models or numerical models scenarios?
Line 357 : sentence too long and not clear. It seems without sense.
Line 365: efficiency in term of?
Author Response
Line 10-11: “ The coast-down transition is to long maintain the cooling loop system using the energy stored by the high moment of inertia of the reactor coolant pump unit.”. Sentence is not clear
Reply: “The coast-down transition is to long maintain the cooling loop system using the energy stored by the high moment of inertia of the reactor coolant pump unit.” have been modified to “The coast-down transition process utilizes the energy stored by the high moment of inertia of the reactor coolant pump unit to maintain a normal coolant supply to the core of the cooling loop system for a short period of time.”
Line 11-12: “Due to the moment of inertia, the unit requires a high reliability of the nuclear reactor coolant pump and consumes considerable energy…” not clear. Please reformulate.
Reply: “Due to the moment of inertia,” have been modified to “Due to the high inertia moment of the rotor system,”
Line 15-17: the cost down ability? I do not understand
Reply: “improving the coast-down ability of the nuclear reactor coolant pump” have been modified to “and to prolong the coast-down time of the reactor coolant pump”
Line 22: sentence too strong: “coolant pump is the key”. Reference to this statement?
Reply: “Nuclear reactor coolant pump is the key equipment in the primary loop cooling system of nuclear power plant.” have been modified to “Nuclear reactor coolant pump is the only rotating equipment in the primary loop cooling system and the “heart” of the nuclear power plant.”
Line 24: In a short time: seconds, minutes?
Reply: “In a short time” have been modified to “In a short period of time,” about 10 seconds.
Line 50 of the impeller
Reply: “The geometrical parameters of impeller are one of the critical factors impacting pump performance.” have been modified to “Geometric parameters of impeller are one of the main factors affecting pump performance.”
Line 57 CFD?
Reply: “CFD approach” have been modified to “CFD numerical simulation method”
Line 69: the geometric parameters, the efficiency and the hydraulic head.
Reply: “between the geometric parameters and the efficiency and the head” have been modified to “between impeller geometric parameters and efficiency, impeller geometric parameters and head.”
Line 90-91: the equation 1 give Et, but at line 91 the equation of Et is different. It is right?
Reply: Equation 1 represents an equation of equal potential energy and kinetic energy.
Fig 2 : caption: describe symbols used. Give separate explanation for the two scheme.
Reply: “In Fig.2, γ denotes outlet inclination of impeller,°; β2 denotes outlet angle of impeller,°; φ denotes wrap angle of impeller,°; Z denotes number of blade of impeller; D2 denotes outlet diameter of impeller,mm; b2 denotes outlet width of impeller,mm; Dj denotes inlet diameter, mm.” have been added. Fig.1 and 2 are schematic descriptions of the pump structure, not two schemes.
Equation 3: what are x*ij, xij, x, n, P?
Equation 4: what is Sj also?
I suggest to remove equation 3 and 4 and give only eq. 5.
Reply: Delete equations 3 and 4 as required.
Table 1. What is Y area ratio?
Reply: “ Area ratio Y represents the ratio of the impeller outlet area to the volute throat area.” have been added.
Table 2: as the coefficient are “adimesional”, they have same accuracy, so they have the same number of decimal digits
Reply: The valid numbers after decimal points in Table 2 are unified.
Line 161, 204 punctuation (or upper case) errors
Reply: This error has been corrected.
Figure 5: measurements unit for T1?
Reply: Dimensionless conversion is carried out with the optimized coast-down time of the pump as a cycle.
Line 312-313: please reformulate in a clear sentence.
Reply: This sentence has been reformulated.
Line 313: “hydraulic head”
Reply:
Line 318: acquisition accuracy of 0.5->missing measurement unit
Reply: It should be “acquisition accuracy of 0.5 grade”, which means that the basic error of the instrument is 0.5%.
Line 324 precision is 0.2, missing measurement unit.
Reply: It should be “the precision is 0.2%,”
Line 347-348: Against energy of unit moment of inertia is stored to keep nuclear reactor coolant pump running 347 in the coast-down transition->sentence is not clear and need to be rewritten in an appropriate English.
Reply: “Against energy of unit moment of inertia is stored to keep nuclear reactor coolant pump running in the coast-down transition.” have been modified to “The inertia moment energy stored in the rotor system is utilized to keep nuclear reactor coolant pump running in the coast-down transition.”
Line 348:Given that the additional losses of the->loses of what?
Reply: Additional loss refers to the reduction of conversion efficiency from rotor kinetic energy to fluid kinetic energy under off-design conditions. Compared with the design conditions, there are additional energy losses such as secondary reflux and eddy current inside the pump.
Line 354-356: sentence too long and not clear. Upper case/lower case and punctuation errors.
Reply: This sentence has been reformulated.
Line 357: what are the external characteristic results? Also they are not “calculation models”, may be they are numerical models or numerical models scenarios?
Reply: It should be “hydraulic performances results”, In pump terminology, external characteristics refer to hydraulic performance, such as head and efficiency.
Line 357 : sentence too long and not clear. It seems without sense.
Reply:This sentence has been reformulated.
Line 365: efficiency in term of?
Reply: It have been modified to “With the efficiency of pump as the target”

Reviewer 2 Report
Language and style are still not suitable.
Author Response
Language and style are still not suitable.
Reply: Some grammatical errors have been corrected.
See the red marking section of the manuscript for details.
Round 3
Reviewer 1 Report
The paper is now ready for publication
Reviewer 2 Report
No more remarks